# AN EFFICIENT FRAMEWORK FOR LEARNING SENTENCE REPRESENTATIONS

**Lajanugen Logeswaran**[*] **& Honglak Lee**[†*]
[*]University of Michigan, Ann Arbor, MI, USA
[†]Google Brain, Mountain View, CA, USA
{llajan,honglak}@umich.edu,honglak@google.com

## ABSTRACT

In this work we propose a simple and efficient framework for learning sentence representations from unlabelled data. Drawing inspiration from the distributional hypothesis and recent work on learning sentence representations, we reformulate the problem of predicting the context in which a sentence appears as a classification problem. Given a sentence and the context in which it appears, a classifier distinguishes context sentences from other contrastive sentences based on their vector representations. This allows us to efficiently learn different types of encoding functions, and we show that the model learns high-quality sentence representations. We demonstrate that our sentence representations outperform state-of-the-art unsupervised and supervised representation learning methods on several downstream NLP tasks that involve understanding sentence semantics while achieving an order of magnitude speedup in training time.

## 1 INTRODUCTION

Methods for learning meaningful representations of data have received widespread attention in recent years. It has become common practice to exploit these representations trained on large corpora for downstream tasks since they capture a lot of prior knowlege about the domain of interest and lead to improved performance. This is especially attractive in a transfer learning setting where only a small amount of labelled data is available for supervision.

Unsupervised learning allows us to learn useful representations from large unlabelled corpora. The idea of self-supervision has recently become popular where representations are learned by designing learning objectives that exploit labels that are freely available with the data. Tasks such as predicting the relative spatial location of nearby image patches (Doersch et al., 2015), inpainting (Pathak et al., 2016) and solving image jigsaw puzzles (Noroozi & Favaro, 2016) have been successfully used for learning visual feature representations. In the language domain, the distributional hypothesis has been integral in the development of learning methods for obtaining semantic vector representations of words (Mikolov et al., 2013b). This is the assumption that the meaning of a word is characterized by the word-contexts in which it appears. Neural approaches based on this assumption have been successful at learning high quality representations from large text corpora.

Recent methods have applied similar ideas for learning sentence representations (Kiros et al., 2015; Hill et al., 2016; Gan et al., 2016). These are encoder-decoder models that learn to predict/reconstruct the context sentences of a given sentence. Despite their success, several modelling issues exist in these methods. There are numerous ways of expressing an idea in the form of a sentence. The ideal semantic representation is insensitive to the form in which meaning is expressed. Existing models are trained to reconstruct the surface form of a sentence, which forces the model to not only predict its semantics, but aspects that are irrelevant to the meaning of the sentence as well.

The other problem associated with these models is computational cost. These methods have a word level reconstruction objective that involves sequentially decoding the words of target sentences. Training with an output softmax layer over the entire vocabulary is a significant source of slowdown in the training process. This further limits the size of the vocabulary and the model (Variations of the softmax layer such as hierarchical softmax (Mnih & Hinton, 2009), sampling based softmax (Jean et al., 2014) and sub-word representations (Sennrich et al., 2015) can help alleviate this issue).

We circumvent these problems by proposing an objective that operates directly in the space of sentence embeddings. The generation objective is replaced by a discriminative approximation where the model attempts to identify the embedding of a correct target sentence given a set of sentence candidates. In this context, we interpret the 'meaning' of a sentence as the information in a sentence that allows it to predict and be predictable from the information in context sentences. We name our approach **quick thoughts (QT)**, to mean efficient learning of thought vectors.

Our key contributions in this work are the following:

- We propose a simple and general framework for learning sentence representations efficiently. We train widely used encoder architectures an order of magnitude faster than previous methods, achieving better performance at the same time.
- We establish a new state-of-the-art for unsupervised sentence representation learning methods across several downstream tasks that involve understanding sentence semantics.

The pre-trained encoders will be made publicly available.

## 2 RELATED WORK

We discuss prior approaches to learning sentence representations from labelled and unlabelled data.

**Learning from Unlabelled corpora.** Le & Mikolov (2014) proposed the paragraph vector (PV) model to embed variable-length text. Models are trained to predict a word given it's context or words appearing in a small window based on a vector representation of the source document. Unlike most other methods, in this work sentences are considered as atomic units instead of as a compositional function of its words.

Encoder-decoder models have been successful at learning semantic representations. Kiros et al. (2015) proposed the skip-thought vectors model, which consists of an encoder RNN that produces a vector representation of the source sentence and a decoder RNN that sequentially predicts the words of adjacent sentences. Drawing inspiration from this model, Gan et al. (2016) explore the use of convolutional neural network (CNN) encoders. The base model uses a CNN encoder and reconstructs the input sentence as well as neighboring sentences using an RNN. They also consider a hierarchical version of the model which sequentially reconstructs sentences within a larger context.

Autoencoder models have been explored for representation learning in a wide variety of data domains. An advantage of autoencoders over context prediction models is that they do not require ordered sentences for learning. Socher et al. (2011) proposed recursive autoencoders which encode an input sentence using a recursive encoder and a decoder reconstructs the hidden states of the encoder. Hill et al. (2016) considered a de-noising autoencoder model (SDAE) where noise is introduced in a sentence by deleting words and swapping bigrams and the decoder is required to reconstruct the original sentence. Bowman et al. (2015) proposed a generative model of sentences based on a variational autoencoder.

Kenter et al. (2016) learn bag-of-words (BoW) representations of sentences by considering a conceptually similar task of identifying context sentences from candidates and evaluate their representations on sentence similarity tasks. Hill et al. (2016) introduced the FastSent model which uses a BoW representation of the input sentence and predicts the words appearing in context (and optionally, the source) sentences. The model is trained to predict whether a word appears in the target sentences. Arora et al. (2016) consider a weighted BoW model followed by simple post-processing and show that it performs better than BoW models trained on paraphrase data.

Jernite et al. (2017) use paragraph level coherence as a learning signal to learn representations. The following related task is considered in their work. Given the first three sentences of a paragraph, choose the next sentence from five sentences later in the paragraph. Related to our objective is the local coherence model of Li & Hovy (2014) where a binary classifier is trained to identify coherent/incoherent sentence windows. In contrast, we only encourage observed contexts to be more plausible than contrastive ones and formulate it as a multi-class classification problem. We experimentally found that this relaxed constraint helps learn better representations.

Encoder-decoder based sequence models are known to work well, but they are slow to train on large amounts of data. On the other hand, bag-of-words models train efficiently by ignoring word order.

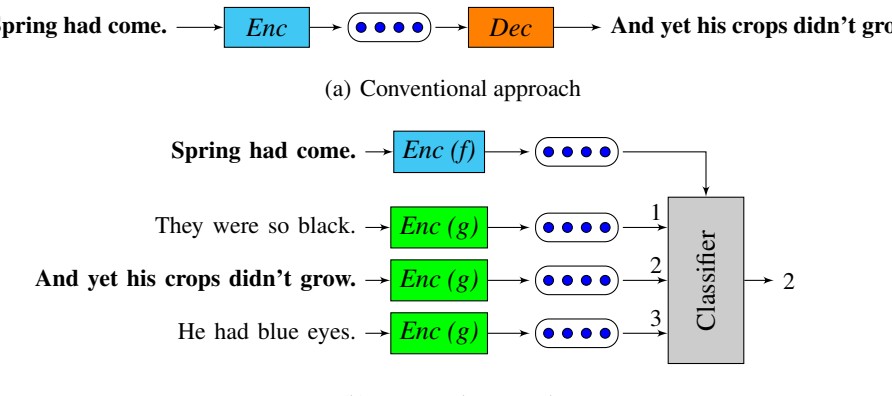

(a) Conventional approach

(b) Proposed approach

Figure 1: Overview. (a) The approach adopted by most prior work where given an input sentence the model attempts to generate a context sentence. (b) Our approach replaces the decoder with a classifier which chooses the target sentence from a set of candidate sentences.

We incorporate the best of both worlds by retaining flexibility of the encoder architecture, while still being able to to train efficiently.

**Structured Resources.** There have been attempts to use labeled/structured data to learn sentence representations. Hill et al. (2016) learn to map words to their dictionary definitions using a max margin loss that encourages the encoded representation of a definition to be similar to the corresponding word. Wieting et al. (2015) and Wieting & Gimpel (2017) use paraphrase data to learn an encoder that maps synonymous phrases to similar embeddings using a margin loss. Hermann & Blunsom (2013) consider a similar objective of minimizing the inner product between paired sentences in different languages. Wieting et al. (2017) explore the use of machine translation to obtain more paraphrase data via back-translation and use it for learning paraphrastic embeddings.

Conneau et al. (2017) consider the supervised task of Natural language inference (NLI) as a means of learning generic sentence representations. The task involves identifying one of three relationships between two given sentences - entailment, neutral and contradiction. The training strategy consists of learning a classifier on top of the embeddings of the input pair of sentences. The authors show that sentence encoders trained for this task perform strongly on downstream transfer tasks.

## 3 PROPOSED FRAMEWORK

The distributional hypothesis has been operationalized by prior work in different ways. A common approach is illustrated in Figure 1(a), where an encoding function computes a vector representation of an input sentence, and then a decoding function attempts to generate the words of a target sentence conditioned on this representation. In the skip-thought model, the target sentences are those that appear in the neighborhood of the input sentence. There have been variations on the decoder such as autoencoder models which predict the input sentence instead of neighboring sentences (Hill et al., 2016) and predicting properties of a window of words in the input sentence (Le & Mikolov, 2014).

Instead of training a model to reconstruct the surface form of the input sentence or its neighbors, we take the following approach. Use the meaning of the current sentence to predict the meanings of adjacent sentences, where meaning is represented by an embedding of the sentence computed from an encoding function. Despite the simplicity of the modeling approach, we show that it facilitates learning rich representations.

Our approach is illustrated in figure 1(b). Given an input sentence, it is encoded as before using some function. But instead of generating the target sentence, the model chooses the correct target sentence from a set of candidate sentences. Viewing generation as choosing a sentence from all possible sentences, this can be seen as a discriminative approximation to the generation problem.

A key difference between these two approaches is that in figure 1(b), the model can choose to ignore aspects of the sentence that are irrelevant in constructing a semantic embedding space. Loss

functions defined in a feature space as opposed to the raw data space have been found to be more attractive in recent work for similar reasons (Larsen et al., 2015; Pathak et al., 2017).

Formally described, let $f$ and $g$ be parametrized functions that take a sentence as input and encode it into a fixed length vector. Let $s$ be a given sentence. Let $S_{ctxt}$ be the set of sentences appearing in the context of $s$ (for a particular context size) in the training data. Let $S_{cand}$ be the set of candidate sentences considered for a given context sentence $s_{ctxt} \in S_{ctxt}$. In other words, $S_{cand}$ contains a valid context sentence $s_{ctxt}$ (ground truth) and many other non-context sentences, and is used for the classification objective as described below.

For a given sentence position in the context of $s$ (e.g., the next sentence), the probability that a candidate sentence $s_{cand} \in S_{cand}$ is the correct sentence (i.e., appearing in the context of $s$) for that position is given by

$$p(s_{cand}|s, S_{cand}) = \frac{\exp[c(f(s), g(s_{cand}))]}{\sum_{s' \in S_{cand}} \exp[c(f(s), g(s'))]} \tag{1}$$

where $c$ is a scoring function/classifier.

The training objective maximizes the probability of identifying the correct context sentences for each sentence in the training data $D$.

$$\sum_{s \in D} \sum_{s_{ctxt} \in S_{ctxt}} \log p(s_{ctxt}|s, S_{cand}) \tag{2}$$

The modeling approach encapsulates the Skip-gram approach of Mikolov et al. (2013b) when words play the role of sentences. In this case the encoding functions are simple lookup tables considering words to be atomic units, and the training objective maximizes the similarity between the source word and a target word in its context given a set of negative samples.

Alternatively, we considered an objective function similar to the negative sampling approach of Mikolov et al. (2013b). This takes the form of a binary classifier which takes a sentence window as input and classifies them as plausible and implausible context windows. We found objective (2) to work better, presumably due to the relaxed constraint it imposes. Instead of requiring context windows to be classified as positive/negative, it only requires ground-truth contexts to be more plausible than contrastive contexts. This objective also performed empirically better than a max-margin loss.

In our experiments, $c$ is simply defined to be an inner product $c(u, v) = u^T v$. This was motivated by considering pathological solutions where the model learns poor sentence encoders and a rich classifier to compensate for it. This is undesirable since the classifier will be discarded and only the sentence encoders will be used to extract features for downstream tasks. Minimizing the number of parameters in the classifier encourages the encoders to learn disentangled and useful representations.

We consider $f, g$ to have different parameters, although they were motivated from the perspective of modeling sentence meaning. Another motivation comes from word representation learning methods which use different sets of input and output parameters. Parameter sharing is further not a significant concern since these models are trained on large corpora. At test time, for a given sentence $s$ we consider its representation to be the concatenation of the outputs of the two encoders $[f(s)\ g(s)]$.

Our framework allows flexible encoding functions to be used. We use RNNs as $f$ and $g$ as they have been widely used in recent sentence representation learning methods. The words of the sentence are sequentially fed as input to the RNN and the final hidden state is interpreted as a representation of the sentence. We use gated recurrent units (GRU) (Chung et al., 2015) as the RNN cell similar to Kiros et al. (2015).

## 4 EXPERIMENTAL RESULTS

### 4.1 EVALUATING SENTENCE EMBEDDINGS

We evaluate our sentence representations by using them as feature representations for downstream NLP tasks. Alternative fine-grained evaluation tasks such as identifying word appearance and word order were proposed in Adi et al. (2017). Although this provides some useful insight about the representations, these tasks focus on the syntactic aspects of a sentence. We are more interested in

assessing how well representations capture sentence semantics. Although limitations of these evaluations have been pointed out, we stick to the traditional approach of evaluating using downstream tasks.

## 4.2 DATA

Models were trained on the 7000 novels of the BookCorpus dataset (Kiros et al., 2015). The dataset consists of about 45M ordered sentences. We also consider a larger corpus for training: the UMBC corpus (Han et al., 2013), a dataset of 100M web pages crawled from the internet, preprocessed and tokenized into paragraphs. The dataset has 129M sentences, about three times larger than BookCorpus. For models trained from scratch, we used case-sensitive vocabularies of sizes 50k and 100k for the two datasets respectively.

## 4.3 TRAINING

A minibatch is constructed using a contiguous sets of sentences in the corpus. For each sentence, all the sentences in the minibatch are considered to be the candidate pool $S_{cand}$ of sentences for classification. This simple scheme for picking contrastive sentences performed as well as other schemes such as random sampling and picking nearest neighbors of the input sentence. Hyperparameters including batch size, learning rate, prediction context size were obtained using prediction accuracies (accuracy of predicting context sentences) on the validation set. A context size of 3 was used, i.e., predicting the previous and next sentences given the current sentence. We used a batch size of 400 and learning rate of 5e-4 with the Adam optimizer for all experiments. All our RNN-based models are single-layered and use GRU cells. Weights of the GRU are initialized using uniform Xavier initialization and gate biases are initialized to 1. Word embeddings are initialized from $U[-0.1, 0.1]$.

## 4.4 EVALUATION

**Tasks** We evaluate the sentence representations on tasks that require understanding sentence semantics. The following classification benchmarks are commonly used: movie review sentiment (MR) (Pang & Lee, 2005), product reviews (CR) (Hu & Liu, 2004), subjectivity classification (SUBJ) (Pang & Lee, 2004), opinion polarity (MPQA) (Wiebe et al., 2005), question type classification (TREC) (Voorhees & Buckland, 2003) and paraphrase identification (MSRP) (Dolan et al., 2004). The semantic relatedness task on the SICK dataset (Marelli et al., 2014) involves predicting relatedness scores for a given pair of sentences that correlate well with human judgements.

The MR, CR, SUBJ, MPQA tasks are binary classification tasks. 10-fold cross validation is used in reporting test performance for these tasks. The other tasks come with train/dev/test splits and the dev set is used for choosing the regularization parameter. We follow the evaluation scheme of Kiros et al. (2015) where feature representations of sentences are obtained from the trained encoders and a logistic/softmax classifier is trained on top of the embeddings for each task while keeping the sentence embeddings fixed. Kiros et al.'s scripts are used for evaluation.

### 4.4.1 COMPARISON AGAINST UNSUPERVISED METHODS

Table 1 compares our work against representations from prior methods that learn from unlabelled data. The dimensionality of sentence representations and training time are also indicated. For our RNN based encoder we consider variations that are analogous to the skip-thought model. The uni-QT model uses uni-directional RNNs as the sentence encoders $f$ and $g$. In the bi-QT model, the concatenation of the final hidden states of two RNNs represent $f$ and $g$, each processing the sentence in a different (forward/backward) direction. The combine-QT model concatenates the representations (at test time) learned by the uni-QT and bi-QT models.

**Models trained from scratch on BookCorpus**. While the FastSent model is efficient to train (training time of 2h), this efficiency stems from using a bag-of-words encoder. Bag of words provides a strong baseline because of its ability to preserves word identity information. However, the model performs poorly compared to most of the other methods. Bag-of-words is also conceptually less attractive as a representation scheme since it ignores word order, which is a key aspect of meaning.

The de-noising autoencoder (SDAE) performs strongly on the paraphrase detection task (MSRP). This is attributable to the reconstruction (autoencoding) loss which encourages word identity and

| Model | Dim | Training time (h) | MR | CR | SUBJ | MPQA | TREC | MSRP (Acc) | MSRP (F1) | SICK r | SICK ρ | SICK MSE |
|---|---|---|---|---|---|---|---|---|---|---|---|---|
| GloVe BoW | 300 | - | 78.1 | 80.4 | 91.9 | 87.8 | 85.2 | 72.5 | 81.1 | 0.764 | 0.687 | 0.425 |
| *Trained from scratch on BookCorpus data* | | | | | | | | | | | | |
| SDAE | 2400 | 192 | 67.6 | 74.0 | 89.3 | 81.3 | 77.6 | **76.4** | 83.4 | N/A | N/A | N/A |
| FastSent | <500 | 2* | 71.8 | 78.4 | 88.7 | 81.5 | 76.8 | 72.2 | 80.3 | N/A | N/A | N/A |
| ParagraphVec | <500 | 4* | 61.5 | 68.6 | 76.4 | 78.1 | 55.8 | 73.6 | 81.9 | N/A | N/A | N/A |
| uni-skip | 2400 | 336 | 75.5 | 79.3 | 92.1 | 86.9 | 91.4 | 73.0 | 81.9 | 0.848 | 0.778 | 0.287 |
| bi-skip | 2400 | 336 | 73.9 | 77.9 | 92.5 | 83.3 | 89.4 | 71.2 | 81.2 | 0.841 | 0.770 | 0.300 |
| combine-skip | 4800 | 336† | 76.5 | 80.1 | **93.6** | 87.1 | **92.2** | 73.0 | 82.0 | 0.858 | 0.792 | 0.269 |
| combine-cnn | 4800 | - | 77.2 | 80.9 | 93.1 | **89.1** | 91.8 | 75.5 | 82.6 | 0.853 | 0.789 | 0.279 |
| *uni-QT* | 2400 | 11 | 77.2 | 82.8 | 92.4 | 87.2 | 90.6 | 74.7 | 82.7 | 0.844 | 0.778 | 0.293 |
| *bi-QT* | 2400 | 9 | 77.0 | 83.5 | 92.3 | 87.5 | 89.4 | 74.8 | 82.9 | 0.855 | 0.787 | 0.274 |
| *combine-QT* | 4800 | 11† | **78.2** | **84.4** | 93.3 | 88.0 | 90.8 | 76.2 | **83.5** | **0.860** | **0.796** | **0.267** |
| *Trained on BookCorpus, pre-trained word vectors are used* | | | | | | | | | | | | |
| combine-cnn | 4800 | - | 77.8 | 82.1 | 93.6 | **89.4** | 92.6 | 76.5 | 83.8 | 0.862 | 0.798 | 0.267 |
| *MC-QT* | 4800 | 11 | **80.4** | **85.2** | **93.9** | **89.4** | **92.8** | **76.9** | **84.0** | **0.868** | **0.801** | **0.256** |
| *Trained on (BookCorpus + UMBC) data, from scratch and using pre-trained word vectors* | | | | | | | | | | | | |
| *combine-QT* | 4800 | 28 | 81.3 | 84.5 | 94.6 | 89.5 | 92.4 | 75.9 | 83.3 | 0.871 | 0.807 | 0.247 |
| *MC-QT* | 4800 | 28 | 82.4 | 86.0 | 94.8 | 90.2 | 92.4 | 76.9 | 84.0 | 0.874 | 0.811 | 0.243 |

Table 1: Comparison of sentence representations on downstream tasks. The baseline methods are GloVe bag-of-words representation, De-noising auto-encoders and FastSent from Hill et al. (2016), the paragraph vector distributed memory model (Le & Mikolov, 2014), skip-thought vectors (Kiros et al., 2015) and the CNN model of Gan et al. (2016). Training times indicated using * refers to CPU trained models and † assumes concatenated representations are trained independently. Performance figures for SDAE, FastSent and ParagraphVec were obtained from Hill et al. (2016). Higher numbers are better in all columns except for the last (MSE). The table is divided into different sections. The bold-face numbers indicate the best performance values among models in the current and all previous sections. Best overall values in each column are underlined.

order information to be encoded in the representation. However, it fails to perform well in other tasks that require higher level sentence understanding and is also inefficient to train.

Our uni/bi/combine-QT variations perform comparably (and in most cases, better) to the skip-thought model and the CNN-based variation of Gan et al. (2016) in all tasks despite requiring much less training time. Since these models were trained from scratch, this also shows that the model learns good word representations as well.

**MultiChannel-QT**. Next, we consider using pre-trained word vectors to train the model. The MultiChannel-QT model (MC-QT) is defined as the concatenation of two bi-directional RNNs. One of these uses fixed pre-trained word embeddings coming from a large vocabulary ($\sim$ 3M) as input. While the other uses tunable word embeddings trained from scratch (from a smaller vocabulary $\sim$ 50k). This model was inspired by the multi-channel CNN model of Kim (2014) which considered two sets of embeddings. With different input representations, the two models discover less redundant features, as opposed to the uni and bi variations suggested in Kiros et al. (2015). We use GloVe vectors (Pennington et al., 2014) as pre-trained word embeddings. The MC-QT model outperforms all previous methods, including the variation of Gan et al. (2016) which uses pre-trained word embeddings.

**UMBC data.** Because our framework is efficient to train, we also experimented on a larger dataset of documents. Results for models trained on BookCorpus and UMBC corpus pooled together ($\sim$ 174M sentences) are shown at the bottom of the table. We observe strict improvements on a majority of the tasks compared to our BookCorpus models. This shows that we can exploit huge corpora to obtain better models while keeping the training time practically feasible.

**Computational efficiency.** Our models are implemented in Tensorflow. Experiments were performed using cuda 8.0 and cuDNN 6.0 libraries on a GTX Titan X GPU. Our best BookCorpus model (MC-QT) trains in just under 11hrs (On both the Titan X and GTX 1080). Training time for the skip-thoughts model is mentioned as 2 weeks in Kiros et al. (2015) and a more recent Tensorflow

implementation[1] reports a training time of 9 days on a GTX 1080. On the augmented dataset our models take about a day to train, and we observe monotonic improvements in all tasks except the TREC task. Our framework allows training with much larger vocabulary sizes than most previous models. Our approach is also memory efficient. The paragraph vector model has a big memory footprint since it has to store vectors of documents used for training. Softmax computations over the vocabulary in the skip-thought and other models with word-level reconstruction objectives incur heavy memory consumption. Our RNN based implementation (with the indicated hyperparamters and batch size) fits within 3GB of GPU memory, a majority of it consumed by the word embeddings.

### 4.4.2 COMPARISON AGAINST SUPERVISED METHODS

| Model | MR | CR | SUBJ | MPQA | SST | TREC | MSRP | | SICK |
|---|---|---|---|---|---|---|---|---|---|
| CaptionRep | 61.9 | 69.3 | 77.4 | 70.8 | - | 72.2 | - | - | - |
| DictRep | 76.7 | 78.7 | 90.7 | 87.2 | - | 81.0 | 68.4 | 76.8 | - |
| NMT En-to-Fr | 64.7 | 70.1 | 84.9 | 81.5 | - | 82.8 | - | - | - |
| InferSent | 81.1 | **86.3** | 92.4 | **90.2** | 84.6 | 88.2 | 76.2 | 83.1 | **0.884** |
| MC-QT | **82.4** | 86.0 | **94.8** | **90.2** | **87.6** | **92.4** | **76.9** | **84.0** | 0.874 |

Table 2: Comparison against supervised representation learning methods on downstream tasks.

| Model | MR | CR | SUBJ | MPQA | SST | TREC | MSRP | | SICK |
|---|---|---|---|---|---|---|---|---|---|
| Ensemble | 82.7 | **86.7** | **95.5** | 90.3 | **88.2** | 93.4 | 78.5 | 85.1 | **0.881** |
| *Task specific methods* | | | | | | | | | |
| AdaSent | **83.1** | 86.3 | **95.5** | **93.3** | - | 92.4 | - | - | - |
| CNN | 81.5 | 85.0 | 93.4 | 89.6 | 88.1 | **93.6** | - | - | - |
| TF-KLD | - | - | - | - | - | - | **80.4** | **85.9** | - |
| DT-LSTM | - | - | - | - | - | - | - | - | 0.868 |

Table 3: Comparison against task-specific supervised models. The models are AdaSent (Zhao et al., 2015), CNN (Kim, 2014), TF-KLD (Ji & Eisenstein, 2013) and Dependency-Tree LSTM (Tai et al., 2015). Note that our performance values correspond to a linear classifier trained on fixed pre-trained embeddings, while the task-specific methods are tuned end-to-end.

Table 2 compares our approach against methods that learn from labelled/structured data. The CaptionRep, DictRep and NMT models are from Hill et al. (2016) which are trained respectively on the tasks of matching images and captions, mapping words to their dictionary definitions and machine translation. The InferSent model of Conneau et al. (2017) is trained on the NLI task. In addition to the benchmarks considered before, we additionally also include the sentiment analysis binary classification task on Stanford Sentiment Treebank (SST) (Socher et al., 2013).

The Infersent model has strong performance on the tasks. Our multichannel model trained on the (BookCorpus + UMBC) data outperforms InferSent in most of the tasks, with most significant margins in the SST and TREC tasks. Infersent is strong in the SICK task presumably due to the following reasons. The model gets to observes near paraphrases (entailment relationship) and sentences that are not-paraphrases (contradiction relationship) at training time. Furthermore, it considers difference features ($|u - v|$) and multiplicative features ($u * v$) of the input pair of sentences $u, v$ during training. This is identical to the feature transformations used in the SICK evaluation as well.

**Ensemble** We consider ensembling to exploit the strengths of different types of encoders. Since our models are efficient to train, we are able to feasibly train many models. We consider a subset of the following model variations for the ensemble.

- Model type - Uni/Bi-directional RNN
- Word embeddings - Trained from scratch/Pre-trained
- Dataset - BookCorpus/UMBC

---

[1] https://github.com/tensorflow/models/tree/master/research/skip_thoughts

| COCO Retrieval | | | | | | | | |
|---|---|---|---|---|---|---|---|---|
| | Image Annotation | | | | Image Search | | | |
| **Model** | **R@1** | **R@5** | **R@10** | **Med** r | **R@1** | **R@5** | **R@10** | **Med** r |
| *Pre-trained unsupervised sentence representations* | | | | | | | | |
| Combine-skip | 33.8 | 67.7 | 82.1 | 3 | 25.9 | 60.0 | 74.6 | 4 |
| Combine-cnn | 34.4 | - | - | 3 | 26.6 | - | - | 4 |
| MC-QT | **37.1** | **72.0** | **84.7** | **2** | **27.9** | **63.3** | **78.3** | **3** |
| *Direct supervision of sentence representations* | | | | | | | | |
| DVSA | 38.4 | 69.6 | 80.5 | **1** | 27.4 | 60.2 | 74.8 | 3 |
| GMM+HGLMM | 39.4 | 67.9 | 80.9 | 2 | 25.1 | 59.8 | 76.6 | 4 |
| m-RNN | 41.0 | 73.0 | 83.5 | 2 | 29.0 | 42.2 | 77.0 | 3 |
| Order | **46.7** | **88.9** | - | 2 | **37.9** | **85.9** | - | **2** |

Table 4: Image-caption retrieval. The purely supervised models are respectively from (Karpathy & Fei-Fei, 2015), (Klein et al., 2015), (Mao et al., 2014) and (Vendrov et al., 2015). Best pre-trained representations and best task-specific methods are highlighted.

Models are combined using a weighted average of the predicted log-probabilities of individual models, the weights being normalized validation set performance scores. Results are presented in table 3. Performance of the best purely supervised task-specific methods are shown at the bottom for reference. Note that these numbers are not directly comparable with the unsupervised methods since the sentence embeddings are not fine-tuned. We observe that the ensemble model closely approaches the performance of the best supervised task-specific methods, outperforming them in 3 out of the 8 tasks.

### 4.4.3 IMAGE-SENTENCE RANKING

The image-to-caption and caption-to-image retrieval tasks have been commonly used to evaluate sentence representations in a multi-modal setting. The task requires retrieving an image matching a given text description and vice versa. The evaluation setting is identical to Kiros et al. (2015). Images and captions are represented as vectors. Given a matching image-caption pair $(I, C)$ a scoring function $f$ determines the compatibility of the corresponding vector representations $v_I, v_C$. The scoring function is trained using a margin loss which encourages matching pairs to have higher compatibility than mismatching pairs.

$$\sum_{(I,C)} \sum_{I'} \max\{0, \alpha - f(v_I, v_C) + f(v_I, v_{C'})\} + \sum_{(I,C)} \sum_{C'} \max\{0, \alpha - f(v_I, v_C) + f(v_{I'}, v_C)\} \quad (3)$$

As in prior work, we use VGG-Net features (4096-dimensional) as the image representation. Sentences are represented as vectors using the representation learning method to be evaluated. These representations are held fixed during training. The scoring function used in prior work is $f(x, y) = (Ux)^T (Vy)$ where $U, V$ are projection matrices which project down the image and sentence vectors to the same dimensionality.

The MSCOCO dataset (Lin et al., 2014) has been traditionally used for this task. We use the train/val/test split proposed in Karpathy & Fei-Fei (2015). The training, validation and test sets respectively consist of 113,287, 5000, 5000 images, each annotated with 5 captions. Performance is reported as an average over 5 splits of 1000 image-caption pairs each from the test set. Results are presented in table 3. We outperform previous unsupervised pre-training methods by a significant margin, strictly improving the median retrieval rank for both the annotation and search tasks. We also outperform some of the purely supervised task specific methods by some metrics.

### 4.4.4 NEAREST NEIGHBORS

Our model and the skip-thought model have conceptually similar objective functions. This suggests examining properties of the embedding spaces to better understand how they encode semantics. We consider a nearest neighbor retrieval experiment to compare the embedding spaces. We use a pool of 1M sentences from a Wikipedia dump for this experiment. For a given query sentence, the best neighbor determined by cosine distance in the embedding space is retrieved.

Table 5 shows a random sample of query sentences from the dataset and the corresponding retrieved sentences. These examples show that our retrievals are often more related to the query sentence compared to the skip-thought model. It is interesting to see in the first example that the model identifies a sentence with similar meaning even though the main clause and conditional clause are in a different order. This is in line with our goal of learning representations that are less sensitive to the form in which meaning is expressed.

| | |
|---|---|
| **Query** | Seizures may occur as the glucose falls further . |
| **ST** | It may also occur during an excessively rapid entry into autorotation . |
| **QT** | When brain glucose levels are sufficiently low , seizures may result . |
| **Query** | This evidence was only made public after both enquiries were completed . |
| **ST** | This visa was provided for under Republic Act No . |
| **QT** | These evidence were made public by the United States but concealed the names of sources . |
| **Query** | He kept both medals in a biscuit tin . |
| **ST** | He kept wicket for Middlesex in two first-class cricket matches during the 1891 County Championship . |
| **QT** | He won a three medals at four Winter Olympics . |
| **Query** | The American alligator is the only known natural predator of the panther . |
| **ST** | Their mascot is the panther . |
| **QT** | The American alligator is a fairly large species of crocodilian . |
| **Query** | Several of them died prematurely : Carmen and Toms very young , while Carlos and Pablo both died . |
| **ST** | At the age of 13 , Ahmed Sher died . |
| **QT** | Many of them died in prison . |
| **Query** | Music for " Expo 2068 " originated from the same studio session . |
| **ST** | His 1994 work " Dialogue " was premiered at the Merkin Concert Hall in New York City . |
| **QT** | Music from " Korra " and " Avatar " was also played in concert at the PlayFest festival in Mlaga , Spain in September 2014 . |
| **Query** | Mohammad Ali Jinnah yielded to the demands of refugees from the Indian states of Bihar and Uttar Pradesh , who insisted that Urdu be Pakistan 's official language . |
| **ST** | Georges Charachidz , a historian and linguist of Georgian origin under Dumzil 's tutelage , became a noted specialist of the Caucasian cultures and aided Dumzil in the reconstruction of the Ubykh language . |
| **QT** | Wali Mohammed Wali 's visit thus stimulated the growth and development of Urdu Ghazal in Delhi . |
| **Query** | The PCC , together with the retrosplenial cortex , forms the retrosplenial gyrus . |
| **ST** | The Macro domain from human , macroH2A1.1 , binds an NAD metabolite O-acetyl-ADP-ribose . |
| **QT** | The PCC forms a part of the posteromedial cortex , along with the retrosplenial cortex ( Brodmann areas 29 and 30 ) and precuneus ( located posterior and superior to the PCC ) . |
| **Query** | With the exception of what are known as the Douglas Treaties , negotiated by Sir James Douglas with the native people of the Victoria area , no treaties were signed in British Columbia until 1998 . |
| **ST** | All the assets of the Natal Railway Company , including its locomotive fleet of three , were purchased for the sum of 40,000 by the Natal Colonial Government in 1876 . |
| **QT** | With few exceptions ( the Douglas Treaties of Fort Rupert and southern Vancouver Island ) no treaties were signed . |

Table 5: Nearest neighbors retrieved by the skip-thought model (ST) and our model (QT).

## 5 CONCLUSION

We proposed a framework to learn generic sentence representations efficiently from large unlabelled text corpora. Our simple approach learns richer representations than prior unsupervised and supervised methods, consuming an order of magnitude less training time. We establish a new state-of-

the-art for unsupervised sentence representation learning methods on several downstream tasks. We believe that exploring scalable approaches to learn data representations is key to exploit unlabelled data available in abundance.

ACKNOWLEDGMENTS

This material is based in part upon work supported by IBM 4915012629, NSF CAREER IIS-1453651, and Sloan Research Fellowship. We thank Jongwook Choi, Junhyuk Oh, Kibok Lee, Ruben Villegas, Seunghoon Hong, Xinchen Yan, Yijie Guo and Yuting Zhang for helpful comments and discussions.

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

# A ANALOGY MAKING

In this experiment we compare the ability of our model and skip-thought vectors to reason about analogies in the sentence embedding space. The analogy task has been widely used for evaluating word representations. The task involves answering questions of the type $A : B :: C :?$ where the answer word shares a relationship to word $C$ that is identical to the relationship between words $A$ and $B$. We consider an analogous task at the sentence level and formulate it as a retrieval task where the query vector $v(C) + v(B) - v(A)$ is used to identify the closest sentence vector $v(\hat{D})$ from a pool of candidates. This evaluation favors models that produce meaningful dimensions.

Guu et al. (2017) exploit word analogy datasets to construct sentence tuples with analogical relationships. They mine sentence pairs $(s_1, s_2)$ from the Yelp dataset (Yelp, 2017) which approximately differ by a single word, and use these pairs to construct sentence analogy tuples based on known word analogy tuples. The dataset has 1300 tuples of sentences collected in this fashion. For each sentence tuple we derive 4 questions by considering three of the sentences to form the query vector. The candidate pool for sentence retrieval consists of all sentences in this dataset and 1M other sentences from the Yelp dataset.

Table 6 compares the retrieval performance of our representations and skip-thought vectors on the above task. Results are classified under word-pair categories in the Google and Microsoft word analogy datasets (Mikolov et al., 2013a;c). Our model outperforms skip-thoughts across several categories and has good performance in the family and verb transformation categories .

| Method | Google | | | Microsoft | | | | | |
|---|---|---|---|---|---|---|---|---|---|
| | gram4-superlative | gram3-comparative | family | JJR_JJS | VB_VBD | VBD_VBZ | NN_NNS | JJ_JJS | JJ_JJR |
| Combine-skip | 0.00 | 0.00 | 0.04 | 0.12 | **0.38** | 0.44 | 0.00 | 0.00 | 0.00 |
| MC-QT | **0.04** | **0.06** | **0.34** | **0.18** | 0.34 | **0.52** | **0.06** | **0.06** | **0.08** |

Table 6: Analogy task - Retrieval performance.

| | |
|---|---|
| Q | dr. <person>and his staff are simply amazing ! |
| | dr. <person>and her staff are simply amazing ! ! ! |
| | i had the chicken and my husband had the pulled pork sandwich . |
| A | i had the pulled pork sandwich and my wife had the pulled chicken sandwich. ✓ |
| Q | place looks great inside . |
| | place looked great inside . |
| | the complimentary valet is also a nice touch . |
| A | the complimentary valet was also a nice touch . ✓ |
| Q | i liked the beef better than the chicken . |
| | i like the chicken better than the beef . |
| | i wanted to like this place so badly ! |
| A | i want so badly to like this place ! ! ✓ |
| Q | the egg drop soup is the best . |
| | the egg drop soup is good . |
| | horrible food and worst customer service . |
| A | horrible food and worst customer service . ✗ |

Table 7: Analogy task - Qualitative results. In each table cell the first three sentences form the query and the last sentence is the answer retrieved by the model.

Table 7 shows some qualitative retrieval results. Each row of the table shows three sentences that form the query and the answer identified by the model. The last row shows an example where the model fails. This is a common failure case of both methods where the model assumes that $A$ and $B$ are identical in a question $A : B :: C :?$ and retrieves sentence $C$ as the answer.

These experiments show that the our representations possess better linearity properties. The transformations evaluated here are mostly syntactic transformations involving a few words. It would be interesting to explore other high-level transformations such as switching sentiment polarity and analogical relationships that involve several words in future work.

# B   Semantic Textual Similarity

In this section we assess the representations learned by our encoders on semantic similarity tasks. The STS14 datasets (Agirre et al., 2014) consist of pairs of sentences annotated by humans with similarity scores. Representations are evaluated by measuring the correlation between human judgments and the cosine similarity of vector representations for a given pair of sentences.

We consider two types of encoders trained using our objective - RNN encoders and BoW encoders. Models were trained from scratch on the BookCorpus data. The RNN version is the same as the combine-QT model in Table 1. We describe the BoW encoder training below.

We train a BoW encoder using our training objective. Hyperparameter choices for the embedding size ($\{100, \mathbf{300}, 500\}$), number of contrastive sentences ($\{500, 1000, \mathbf{1500}, 2000\}$) and context size ($\{3, 5, \mathbf{7}\}$) were made based on the validation set (optimal choices highlighted in bold). Training this model on the BookCorpus dataset takes 2 hours on a Titan X GPU. Similar to the RNN encoders, the representation of a sentence is obtained by concatenating the outputs of the input and output sentence encoders.

Table 8 compares different unsupervised representation learning methods trained on the BookCorpus data from scratch. Methods are categorized as sequence models and bag-of-words models. Our RNN-based encoder performs strongly compared to other sequence encoders. Bag-of-words models are known to perform strongly in this task as they are better able to encode word identity information. Our BoW variation performs comparably to prior BoW based models.

| Model | News | Forum | WordNet | Tweets | Images | Headlines | Overall |
|-------|------|-------|---------|--------|--------|-----------|---------|
| *Sequence Models* | | | | | | | |
| SDAE | 0.04 | 0.13 | 0.24 | 0.42 | 0.38 | 0.36 | 0.15 |
| Skip-Thoughts | 0.45 | 0.12 | 0.35 | 0.36 | **0.62** | 0.36 | 0.39 |
| *QT (RNN)* | **0.48** | **0.15** | **0.53** | **0.62** | 0.53 | **0.48** | **0.49** |
| *BoW Models* | | | | | | | |
| CBOW | 0.61 | **0.44** | 0.69 | **0.75** | 0.73 | 0.59 | **0.65** |
| Skipgram | 0.59 | 0.42 | 0.70 | 0.74 | 0.67 | 0.58 | 0.63 |
| FastSent | 0.59 | 0.36 | 0.70 | 0.66 | **0.78** | 0.59 | 0.64 |
| Siamese CBOW | 0.59 | 0.41 | 0.61 | 0.73 | 0.65 | **0.64** | 0.62 |
| *QT (BoW)* | **0.62** | 0.37 | **0.76** | 0.67 | 0.76 | 0.60 | **0.65** |

Table 8: Comparison (Pearson score) of sentence representations on Semantic Textual Similarity (STS14) tasks. SDAE, CBOW, Skipgram and FastSent are from Hill et al. (2016). The other baselines are Skip-Thoughts (Kiros et al., 2015) and Siamese CBOW (Kenter et al., 2016). QT (RNN) and QT (BoW) are our models trained with RNN and BoW encoders, respectively.

## C TRAINING EFFICIENCY

To better assess the training efficiency of our models, we perform the following experiment. We train the same encoder architecture using our objective and the skip-thought (ST) objective and compare the performance after a certain number of hours of training. Since training the ST objective with large embedding sizes takes many days, we consider a lower dimensional sentence encoder for this experiment. We chose the encoder architecture to be a single-layer GRU Recurrent neural net with hidden state size $H = 1000$. The word embedding size was set to $W = 300$ and a vocabulary size of $V = 20,000$ words was used. Both models are initialized randomly from the same distribution. The models are trained on the same data for 1 epoch using the Adam optimizer with learning rate 5e-4 and batch size 400. For the low dimensional model considered, the model trained with our objective and ST objective take 6.5 hrs and 31 hrs, respectively.

The number of parameters for the two objectives are

- Ours: $6H(H + W + 1) + 2VW \approx 19.8M$ parameters
- ST: $9H(H + W + 1) + VW + 2HV \approx 57.7M$ parameters

Only the input side encoder parameters ($\approx 9.9M$ parameters) are used for the evaluation.

The 1000-dimensional sentence embeddings are used for evaluation. Evaluation follows the same protocol as in section 4.4. Figure 2 compares the performance of the two models on downstream tasks after $x$ number of training hours. The speed benefits of our training objective is apparent from these comparisons.

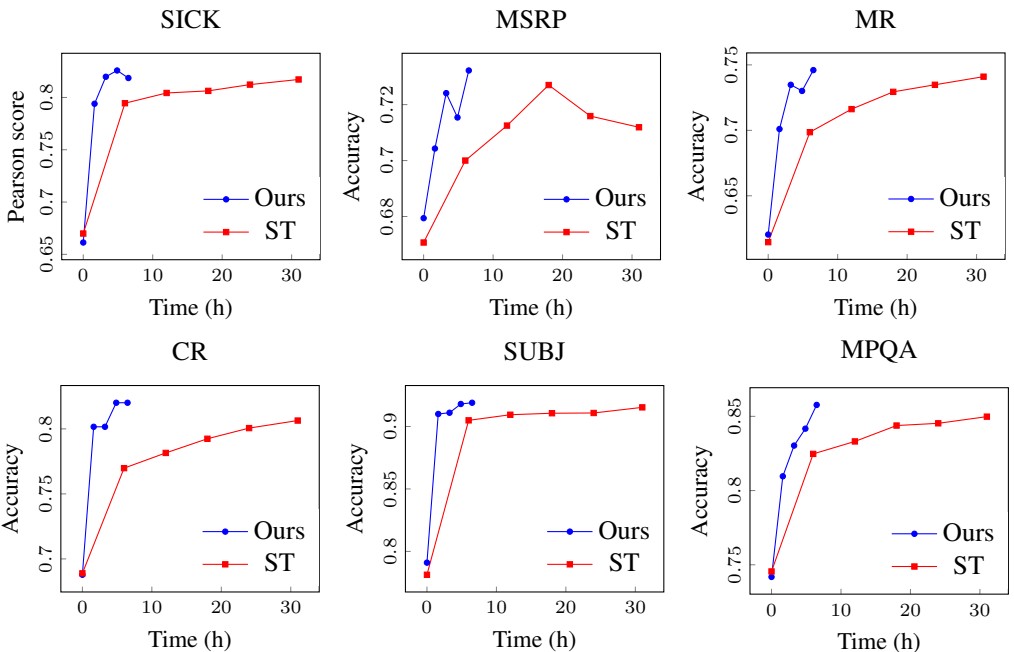

Figure 2: Same encoder architecture trained using our objective and Skip-thought (ST) objective and performance on downstream tasks is compared after a given number of hours.

The overall training speedup observed for our objective is 4.8x. Note that the output encoder was discarded for our model, unlike the experiments in the main text where the representations from the input and output encoders are concatenated. Further speedups can be achieved by training with encoders half the size and concatenating them (This is also parameter efficient).

# D REPRESENTATION SIZE, TRAINING EFFICIENCY AND PERFORMANCE

We explore the trade-off between training efficiency and the quality of representations by varying the representation size. We trained models with different representation sizes and evaluate them on the downstream tasks. The multi-channel model (MC-QT) was used for these experiments. Models were trained on the BookCorpus dataset.

Table 9 shows the training time and the performance corresponding to different embedding sizes. The training times listed here assume that the two component models in MC-QT are trained in parallel. The reported performance is an average over all the classification benchmarks (MSRP, TREC, MR, CR, SUBJ, MPQA). We note that the classifiers trained on top of the embeddings for downstream tasks differ in size for each embedding size. So it is difficult to make any strong conclusions about the quality of embeddings for the different sizes. However, we are able to reduce the embedding size and train the models more efficiently, at the expense of marginal loss in performance in most cases.

The 4800-dimensional Skip-thought model (Kiros et al., 2015) and Combine-CNN model (Gan et al., 2016) achieve mean accuracies of 83.75 and 85.33 respectively. We note that our 1600-dimensional model and 3200-dimensional model are respectively better than these models, in terms of the mean performance across the benchmarks (We acknowledge that the Skip-thought model did not use pre-trained word embeddings). This suggests that high-quality models can be obtained even more efficiently by training lower-dimensional models on large amounts of data using our objective.

| Embedding size | Training time (h) | Performance |
|---|---|---|
| 1600 | 4 | 84.57 |
| 2400 | 4.7 | 85.12 |
| 3200 | 6 | 85.74 |
| 4000 | 7.15 | 86.09 |
| 4800 | 8.5 | 86.43 |
| 5600 | 10.15 | 86.72 |

Table 9: Training time and performance for different embedding sizes. The reported performance is the mean accuracy over the classification benchmarks (MSRP, TREC, MR, CR, SUBJ, MPQA).

