# OpenReview forum: "An efficient framework for learning sentence representations"
_ICLR.cc/2018/Conference — Accept (Poster)_

### Official Review · AnonReviewer3 · 2017-11-19
**Great results, with minor concerns**

**Rating:** 8
**Confidence:** 4

**Review:**

==Update==

I appreciate the response, and continue to recommend acceptance. The evaluation metric used in this paper (SentEval) represents an important open problem in NLP—learning reusable sentence representations—and one of the problems in NLP best suited to presentation at IC*LR*. Because of this, I'm willing to excuse the fact that the paper is only moderately novel, in light of the impressive reported results.

While I would appreciate a direct (same codebase, same data) comparison with some outside baselines, this paper meets or exceeds the standards for rigor that were established by previous published work in the area, and the existing results are sufficient to support some substantial conclusions.

==========

This paper proposes an alternative formulation of Kiros's SkipThought objective for training general-purpose sentence encoder RNNs on unlabeled data. This formulation replaces the decoder in that model with a second encoder, and yields substantial improvements to both speed and model performance (as measured on downstream transfer tasks). The resulting model is, for the first time, reasonably competitive even with models that are trained end-to-end on labeled data for the downstream tasks (despite the requirement, imposed by the evaluation procedure, that only the top layer classifier be trained for the downstream tasks here), and is also competitive with models trained on large labeled datasets like SNLI. The idea is reasonable, the topic is important, and the results are quite strong. I recommend acceptance, with some caveats that I hope can be addressed.

Concerns:

A nearly identical idea to the core idea of this paper was proposed in an arXiv paper this spring, as a commenter below pointed out. That work has been out for long enough that I'd urge you to cite it, but it was not published and it reports results that are far less impressive than yours, so that omission isn't a major problem.

I'd like to see more discussion of how you performed your evaluation on the downstream tasks. Did you use the SentEval tool from Conneau et al., as several related recent papers have? If not, does your evaluation procedure differ from theirs or Kiros's in any meaningful way?

I'm also a bit uncomfortable that the paper doesn't directly compare with any baselines that use the exact same codebase, word representations, hyperparameter tuning procedure, etc.. I would be more comfortable with the results if, for example, the authors compared a low-dimensional version of their model with a low-dimensional version of SkipThought, trained in the *exact* same way, or if they implemented the core of their model within the SkipThought codebase and showed strong results there.

Minor points:

The headers in Table 1 don't make it all that clear which additions (vectors, UMBC) are cumulative with what other additions. This should be an easy fix.

The use of the check-mark as an output in Figure 1 doesn't make much sense, since the task is not binary classification.

"Instead of training a model to reconstruct the surface form of the input sentence or its neighbors, our formulation attempts to focus on the semantic aspects of sentences. The meaning of a sentence is the property that creates bonds between a sequence of sentences and makes it logically flow." – It's hard to pin down exactly what this means, but it sounds like you're making an empirical claim here: semantic information is more important than non-semantic sources of variation (syntactic/lexical/morphological factors) in predicting the flow of a text. Provide some evidence for this, or cut it.

You make a similar claim later in the same section: "In figure 1(a) however, the reconstruction loss forces the model to predict local structural information about target sentences that may be irrelevant to its meaning (e.g., is governed by grammar rules)." This is a testable prediction: Are purely grammatical (non-semantic) variations in sentence form helpful for your task? I'd suspect that they are, at least in some cases, as they might give you clues as to style, dialect, or framing choices that the author made when writing that specific passage.

"Our best BookCorpus model (MC-QT) trains in just under 11hrs, compared to skip-thought model’s training time of 2 weeks." –  If you say this, you need to offer evidence that your model is faster. If you don't use the same hardware and low-level software (i.e., CuDNN), this comparison tells us nearly nothing. The small-scale replication of SkipThought described above should address this issue, if performed.

---

### Official Review · AnonReviewer1 · 2017-11-28
**An elegant and simple alternative to existing methods, but empirical advantages are unclear**

**Rating:** 6
**Confidence:** 5

**Review:**

[REVISION]

Thank you for your clarification. I appreciate the effort and think it has improved the paper. I have updated my score accordingly

======

This paper proposes a new objective for learning SkipThought-style sentence representations from corpora of ordered sentences. The algorithm is much faster than SkipThoughts as it swaps the word-level decoder for a contrastive classification loss.

Comments:

Since one of the key advantages of this method is the speed, I was surprised there was not a more formal comparison of the speed of training different models. For instance, it would be more convincing if two otherwise identical encoders were trained on the same machine on the books corpus with the proposed objective and the skipthoughts decoding objective, and the representations compared after X hours of training. The reported 2 weeks required to train Skipthoughts comes from the paper, but things might be faster now with more up-to-date deep learning libraries etc. If this was what was in fact done, then it's probably just a case of presenting the comparison in a more formal way. I would also lose the sentence "we are able to train many models in the time it takes to train most unsupervised" (see next point for reasons why this is questionable).

It would have been interesting to apply this method with BOW encoders, which should be even faster than RNN-based encoders reported in this paper. The faster BOW models tend to give better performance on cosine-similarity evaluations ( quantifying the nearest-neighbour analysis that the authors use in this paper). Indeed, it would be interesting (although of course not definitive) to see comparison of the proposed algorithm (with BOW and RNN encoders) on cosine sentence similarity evaluations.

The proposed novelty is simple and intuitive, which I think is a strength of the method. However, a simple idea makes overlap with other proposed approaches more likely, and I'd like the author to check through the public comments to ensure that all previous related ideas are noted in this paper.

I think the authors could do more to emphasise what the point is of trying to learn sentence embeddings. An idea of the eventual applications of these embeddings would make it easier to determine, for instance, whether the supervised ensembling method applied here would be applicable in practice. Moreover, many papers have emphasised the limitations of the evaluations used in this paper (although they are still commonly used) so it would be good to acknowledge that it's hard to draw too many conclusions from such numbers. That said, the numbers are comparable Skipthoughts, so it's clear that this method learns representations of comparable quality.

The justification for the proposed algorithm is clear in terms of efficiency, but I don't think it's immediately clear from a semantic / linguistic point of view. The statement "The meaning of a sentence is the property that creates bonds...." seems to have been cooked up to justify the algorithm, not vice versa. I would cut all of that speculation out and focus on empirically verifiable advantages.

The section of image embeddings comes completely out of the blue and is very hard to interpret. I'm still not sure I understand this evaluation (short of looking up the Kiros et al. paper), or how the proposed model is applied to a multi-modal task.

There is much scope to add more structured analysis of the type hinted by the nearest neighbours section. Cherry picked lists don't tell the reader much, but statistics or more general linguistic trends can be found in these neighbours and aggregated, that could be very interesting.

---

### Official Review · AnonReviewer2 · 2017-11-29
**Intuitive model for sentence representations with good performance**

**Rating:** 6
**Confidence:** 4

**Review:**

This paper proposes a framework for unsupervised learning of sentence representations by maximizing a model of the probability of true context sentences relative to random candidate sentences. Unique aspects of this skip-gram style model include separate target- and context-sentence encoders, as well as a dot-product similarity measure between representations. A battery of experiments indicate that the learned representations have comparable or better performance compared to other, more computationally-intensive models.

While the main constituent ideas of this paper are not entirely novel, I think the specific combination of tools has not been explored previously. As such, the novelty of this paper rests in the specific modeling choices and the significance hinges on the good empirical results. For this reason, I believe it is important that additional details regarding the specific architecture and training details be included in the paper. For example, how many layers is the GRU? What type of parameter initialization is used? Releasing source code would help answer these and other questions, but including more details in the paper itself would also be welcome.

Regarding the empirical results, the method does appear to achieve good performance, especially given the compute time. However, the balance between performance and computational complexity is not investigated, and I think such an analysis would add significant value to the paper. For example, I see at least three ways in which performance could be improved at the expense of additional computation: 1) increasing the candidate pool size 2) increasing the corpus size and 3) increasing the embedding size / increasing the encoder capacity. Does the good performance/efficiency reported in the paper depend on achieving a sweet spot among those three hyperparameters?

Overall, the novelty of this paper is fairly low and there is still substantial room for improvement in some of the analysis. On the other hand, I think this paper proposes an intuitive model and demonstrates good performance. I am on the fence, but ultimately I vote to accept this paper for publication.

---

### Public Comment · (anonymous) · 2017-10-28
**Some literature review suggestions**

This article proposed a framework to learn sentence representation and demonstrated some good results.

In terms of the main objective of this task - predicting the next sentence out of a group of sampled sentences has already been proposed multiple times in the NLP community. For example, it appeared earlier this year: https://arxiv.org/abs/1705.00557, and a much earlier work (in 2014) has also used sentence ordering to learn sentence representation: http://web.stanford.edu/~jiweil/paper/emnlp_coherence-v2eh.pdf

I am certain this paper brings unique value and insight into this training objective, and is a much-needed addition to the existing pool of literature. I just hope maybe in a revised version of this paper, the author(s) would reference these previous NLP works.

(To clarify on my intent: I am not related to any of these papers, but would love to see NLP researches get recognized.)

---

> ### Author Response · Authors · 2017-10-31
> **Objective function**
>
> Thank you for your comments. We will include these literature in a revised version of the paper.
>
> Despite similarities in the objective functions, we would like to point out the following key distinctions.
>
> Jernite et al. propose to use paragraph level coherence as a learning signal. The following related task is considered in their paper. Given the first three sentences of a paragraph, they choose the next sentence from five candidate sentences later in the paragraph (Paragraphs of length at least 8 are considered).
> Our objective differs from theirs in the following aspects.
> * This work exploits paragraph level coherence signals for learning, while our work derives motivation from the distributional hypothesis. We don’t restrict ourselves to paragraphs in the data as is done in this work.
> * We consider a large number of candidate sentence choices when predicting a context sentence. This is a discriminative approximation to the generation objective (viewing generation as choosing a sentence from all possible sentences)
> * We use a single input sentence and predict the context sentences surrounding it. Using larger input contexts did not yield any significant empirical benefits.
> Our objective further learns richer representations compared to this work, as evidenced by empirical results.
>
> The local coherence model of Li & Hovy is a feed-forward network which examines a window of sentence embeddings and classifies them as coherent/incoherent (binary classification). We have some discussion about this objective in the paper (section 3). We point out the following key differences between our objective and theirs.
> * Instead of discriminating context windows as plausible/implausible, we encourage observed contexts (in the data) to be more plausible than contrastive (implausible) ones and formulate it as a multi-class classification problem. We experimentally found that this relaxed constraint helps learn better representations.
> * We use a simple scoring function (inner products) in our objective. When using a parameterized classifier, the model has a tendency to learn poor sentence representations and compensate for it using a strong classifier. This is undesirable since the classifier is discarded and only the sentence encoders are used for feature extraction.
>
> Hence, Li & Hovy’s objective is better suited for local coherence modeling than it is for learning sentence representations.

---

### Public Comment · (anonymous) · 2017-10-29
**a related paper**

https://arxiv.org/pdf/1705.00557.pdf

The proposed method in the listed paper is quite close to the one proposed in this submission. I think it'll be good to cite this listed paper and discuss it. (Although I know it is not required to cite arxiv papers.)

(Also, I am not related to the listed arxiv paper, but I'd love to some comprehensive comparisons among existing methods.)

---

> ### Author Response · Authors · 2017-10-31
> **Related paper**
>
> Thank you for your comment. We will include the paper in a revised version. Please see our response to the previous comment regarding the same paper.

---

> > ### Public Comment · (anonymous) · 2017-10-31
> > **Siamese CBOW**
> >
> > Thank you for your reply! Could you also compare your idea with Siamese CBOW? (ACL2016)
> >
> > http://www.aclweb.org/anthology/P16-1089
> >
> > Thanks again!

---

> > > ### Author Response · Authors · 2017-11-03
> > > **Siamese CBOW**
> > >
> > > Sure, Thanks.
> > >
> > > In this paper a conceptually similar task of identifying context sentences from candidate sentences based on their bag-of-words representations is considered. Our approach is more general than this work in the following ways
> > > * Our formulation considers more general scoring functions/classifiers. We found inner products to work best. Using cosine distance as is done in this work led to inferior representations. Cosine distance implicitly requires sentence representations to both lie on the unit ball and be similar (in terms of inner product) to context sentences, which can be a strong constraint. The inner products scoring function only requires the latter.
> > > * This work uses the same set of parameters to encode both input and context sentences, while we consider using different sets of parameters. This helped learn better representations. We briefly discuss this choice in section 3.
> > > * Our formulation also allows the use of more general encoder architectures.
> > >
> > > Also, we discuss more recent bag-of-words methods in the paper.

---

> > > > ### Public Comment · (anonymous) · 2017-11-06
> > > > **Evaluation**
> > > >
> > > > Thanks for your reply.
> > > >
> > > > There is an interesting difference between the evaluation tasks and the evaluation task used in Siamese CBOW.
> > > >
> > > > In Siamese CBOW, they mainly focused on unsupervised evaluation tasks, including STS12, 13, and 14. The similarity of 2 sentences is determined by Cosine-similarity, which matches their training objective. Compared with FastSent which applies the dot-product as training objective in FastSent, Siamese CBOW seems to get better results.
> > > >
> > > > Could you also evaluate your proposed model on unsupervised evaluation tasks, like STS14? It would be good to have a comprehensive evaluation of your model. Thanks!

---

> > > > > ### Author Response · Authors · 2018-01-04
> > > > > **STS14 evaluation**
> > > > >
> > > > > Thank you for your comment.
> > > > >
> > > > > We have included an evaluation of our models on the STS14 task in Appendix C of the supplementary material.
> > > > >
> > > > > We evaluate RNN-based and Bag-of-words encoders trained using our objective on this task. Our RNN-based encoder performs strongly compared to previous sequence encoders. Bag-of-words models are known to perform strongly in this task as they are better able to encode word identity information. Our BoW variation performs comparably (or slightly better) than prior BoW based models such as FastSent and Siamese CBOW.

---

### Public Comment · ~Samuel_R._Bowman1 · 2017-11-01
**Data volume question**

Just out of curiosity, do you have any results on how the quantity of unlabeled training data you use impacts model performance?

---

### Author Response · Authors · 2017-12-20
**Rebuttal**

We thank the reviewers for the helpful comments.

R1, R3: Skip-thoughts training time
We agree that training the model could be faster with current hardware and software libraries. A more recent implementation of the skip-thoughts model was released by Google early this year [1]. This implementation mentions that the model takes 9 days to train on a GTX 1080 GPU. Training our proposed models on a GTX 1080 takes 11 hours. Both implementations are based on Tensorflow. Our experiment used cuda 8.0 and cuDNN 6.0 libraries. This also agrees with the numbers in the paper which were based on experiments using GTX TITAN X.

R1, R3: Training speed comparison
We performed a comparison on the training efficiency of lower-dimensional versions of our model and the skip-thoughts model. The same encoder architecture was trained in identical conditions using our objective and the skip-thoughts objectives and models were evaluated on downstream tasks after a given number of hours. Experimental results are reported in section C of the appendix. The training efficiency of our model compared to the skip-thoughts model is clear from these experiments.

R1: BoW encoders, sentence similarity evaluations
We train BoW encoders using our training objective and evaluate them on textual similarity tasks. Experiments and results are discussed in section B of the appendix. Our RNN-based encoder performs strongly against prior sequence models. Our BoW encoder performs comparably to (or slightly better than) popular BoW representations as well.

R2: Balance between performance and computational complexity
1) Increasing the candidate pool size - We found that RNN encoders are less sensitive to increasing the candidate pool size. Sentences appearing in the context of a given query sentence are natural candidates for the contrastive sentences since they are more likely to be related to the query sentence, and hence make the prediction problem challenging. We observed marginal performance improvements as we added more random choices to the candidate pool.
2) Increasing corpus size - We have experiments in the paper with increased corpus size. We considered the UMBC corpus (which is about 3 times the size of BookCorpus) and show that augmenting the BookCorpus dataset enables us to obtain monotonic improvements on the downstream tasks.
3) Increasing embedding size - We have included experiments on varying the embedding size in section D of the supplementary material. We are able to train bigger and better models at the expense of more training time. The smaller models can be trained more efficiently while still being competitive or better than state-of-the-art higher-dimensional models.
We also plan to release pre-trained models for different representation sizes so that other researchers/practitioners can use the appropriate size depending on the downstream task and the amount of labelled data available.

R1: Point of learning sentence representations
In the vision community it has become common practice to use CNN features (e.g., AlexNet, VGGNet, ResNet, etc.) pre-trained from the large-scale imagenet database for a variety of downstream tasks (e.g., the image-caption experiment in our paper uses pre-trained CNN features as the image embedding). Our overarching goal is to learn analogous high-quality sentence representations in the text domain. The representations can be used as feature vectors for downstream tasks, as we do in the paper. The encoders can also be used for parameter initialization and fine-tuned on data relevant to a particular application. In this respect, we believe that exploring scalable unsupervised learning algorithms for learning ‘universal’ text representations is an important research problem.

R1: Image-caption retrieval experiments
We have updated the description of the image-caption retrieval experiments. We hope the description is more clear now and provides better motivation for the task.

R1: Nearest neighbors
As we discuss in the paper, the query sentences used for the nearest neighbor experiment were chosen randomly and not cherry picked. We hope the cosine similarity experiments quantify the nearest neighbor analysis.

R2: We have added more details about the architecture and training to the paper (sec 4.3).

We will release the source code upon publication.

R3: Evaluation
The evaluation on downstream tasks was performed using the evaluation scripts from Kiros et al. since most of the unsupervised methods we compare against were published either before (Kiros et al., Hill et al.) or about the same time (Gan et al.) the SentEval tool was released.

R1, R2, R3:
We have updated the paper to reflect your comments and concerns. Modifications are highlighted in blue (Omissions not shown). We have added relevant citations pointed out by reviewers and public comments.

References
[1] https://github.com/tensorflow/models/tree/master/research/skip_thoughts

---

### Public Comment · (anonymous) · 2018-02-13
**Reproduce**

It is a very interesting work of utilizing the context information to learn sentence representations efficiently, and the results are quite amazing

I was trying to reproduce your model in pytorch, and I couldn't get comparable results as presented in your paper, maybe I missed some important parts. I was wondering if the authors plan to release the code in the near future, and it will be a big contribution to the community.

Also, I wanted to know if someone has been able to reproduce the results in the paper, and if so, please let me know.

Thanks for the good paper and your hard work on all experiments in your paper.

Cheers,

---

### Decision · Program_Chairs · 2018-01-29
**ICLR 2018 Conference Acceptance Decision**

**Decision:**

Accept (Poster)

**Comment:**

Though the approach is not terribly novel, it is quite effective (as confirmed on a wide range of evaluation tasks). The approach is simple and likely to be useful in applications. The paper is well written.

+ simple and efficient
+ high quality evaluation
+ strong results
- novelty is somewhat limited